# Effects of Tooth Brushing Training, Based on Augmented Reality Using a Smart Toothbrush, on Oral Hygiene Care among People with Intellectual Disability in Korea

**DOI:** 10.3390/healthcare9030348

**Published:** 2021-03-18

**Authors:** Byoungjin Jeon, Jinseok Oh, Sungmin Son

**Affiliations:** 1Department of Occupational Therapy, Kangwon National University, Samcheok 25949, Korea; nomadot@kangwon.ac.kr; 2Department of Emergency Medical Rehabilitation Service, Graduate School of Kangwon National University, Samcheok 25949, Korea; oh486255@naver.com; 3Department of Occupational Therapy, Jeonju Kijeon College, Jeonju 54989, Korea

**Keywords:** augmented reality, intellectual disability, oral hygiene care, residential care facility, smart toothbrush

## Abstract

(1) Purpose: In modern society, augmented reality (AR)-based training using a smart device has emerged as a means of resolving problems with training. Thus, this feasibility study aimed to identify the effects of tooth-brushing training, based on AR using a smart toothbrush, on oral hygiene care among people with an intellectual disability in Korea. (2) Methods: Thirty people with an intellectual disability, residing in a residential care facility, were selected. Tooth-brushing training based on AR, using a smart toothbrush, was applied in the experimental group (*n* = 15), and training using visual material was applied in the control group (*n* = 15). As an assessment of oral hygiene care, the changes in tooth-brushing performance and oral hygiene were measured. (3) Results: There were significant differences in all results after training between the two groups. (4) Conclusions: Tooth-brushing training based on AR using a smart toothbrush is more effective than training using visual material on oral hygiene care among the subjects of this feasibility study. Thus, training based on AR using a smart toothbrush could be applied to people with intellectual disabilities residing in residential care facilities as an individual tool for tooth-brushing training.

## 1. Introduction

Oral hygiene plays an important role in protecting the body from infection, as well as in helping the intraoral structures to maintain proper function to enable chewing and swallowing [1]. However, problems in oral hygiene induce functional problems in intraoral structures and particularly affect masticatory function, thereby diminishing the range and amount of food intake. As a result, they decrease the swallowing-related quality of life, which, in turn, hampers maintenance of health and fitness and deteriorates physical functions. Oral halitosis is significantly associated with dental caries and periodontal diseases, ultimately leading to tooth loss [2]. 

In addition, oral hygiene problems increase oral halitosis, which serves as a negative factor in individuals’ interpersonal relationships and social life. These also affect pronunciation, which increases psychological and emotional problems, such as discouraging people and making them feel isolated and alienated. Thus, proper management of these problems is needed [3]. In particular, these problems in oral hygiene are more evident among people with intellectual disabilities compared to the general public, as they face substantial problems and difficulties with their low intellectual level, cognitive function, learning level, and communication skills [4]. 

Among the various oral hygiene management activities, tooth brushing is an essential activity to actively manage oral hygiene problems. Tooth brushing is a personal hygiene management activity, which is crucial, directly impacting oral hygiene [5]. Furthermore, it helps to prevent and manage oral hygiene problems continuously and actively. In addition, tooth brushing is highly important because it helps to detect oral hygiene problems early and easily resolves them [6]. Owing to this significance, residential care facilities for people with intellectual disabilities continuously administer oral hygiene education to increase residents’ awareness and ability to manage oral hygiene, in addition to tooth-brushing training, to help them form correct tooth-brushing habits to ensure active and continuous management of oral hygiene [5]. 

However, people with intellectual disabilities face substantial problems and difficulties in tooth-brushing activities due to their low intellectual level, cognitive function, learning level, and communication skills. In particular, they are highly dependent on others for tooth brushing and have difficulty learning and performing the correct tooth-brushing technique [4]. As a result, oral hygiene is not properly managed among this group of people, which has led to a high incidence of dental caries and periodontal diseases, with worse conditions compared to the general public [7]. 

Generally, tooth-brushing training and learning is provided by utilizing various visual materials with a general toothbrush in consideration of the characteristics and intellectual level of people with intellectual disabilities [8]. However, since such training is a one-time event and is conducted in a unified format, it is not possible to form correct tooth brushing habits in daily life [9]. Moreover, the visual materials used in such training and learning have limitations in engaging and motivating people with intellectual disabilities to learn tooth-brushing patterns, and difficulties sustaining the effectiveness of learning [10]. 

Moreover, in the case of training, the use of an electronic toothbrush only has the advantages of promoting tooth-brushing movements due to the increasing level of awareness of movements and paying attention to performance due to the provision of electrical stimulation. Therefore, it is limited to learning detailed movements and postures, and increasing the accuracy of tooth-brushing movement [10]. Accordingly, even in training with an electronic toothbrush, in order to learn the proper movement of tooth brushing and form the correct tooth-brushing pattern, additional education processes and training are required. In particular, for people with intellectual disabilities, such education processes and training are required intensively, and the use of various materials, such as visual materials, is needed [8].

De Castilho et al. [10] reported that training using visual materials and electronic toothbrushes has some limitations in the participation and motivation of people with intellectual disabilities, and does not make a substantial contribution to improving the oral environment through tooth-brushing practice. Moreover, they reported that the effects of the training were not persistent afterwards. In addition, such training does not provide systematic feedback on actual tooth-brushing performance, and the instructor conducts it in a uniform format, reducing the learning effect and performance level. Kolawole et al. [11] also reported that this causes difficulties in continuing tooth-brushing performance in daily life after training, and it also makes it difficult to check whether tooth brushing is performed at an appropriate time. As a result, participation in tooth brushing decreases, and in severe cases, tooth brushing is sometimes avoided. 

AR-based training is a type of training in which users engage in customized activities and perform tasks in an AR that is identical to the real world. In addition, it helps the users to repeatedly participate in the activities while having fun [8] and induces the users to face difficult challenges based on their abilities by using the game contents [12]. Owing to these benefits, the use of AR in the training of various activities of daily living and cognitive tasks is increasing. Furthermore, motion sensors equipped in smart devices sensitively recognize the user’s motions, and, based on this, these devices can provide visual and numeric indices of performance and the results in real-time as well as immediate feedback and monitoring of changes in the level of performance and results [13]. 

Studies that applied AR-based training have mostly focused on cognitive tasks such as memory [14,15], concentration [16], and unilateral neglect [17]. AR-based training for activities of daily living has been applied in several studies [12,18]. Various studies have applied AR-based training to tooth brushing. These studies utilized a smart toothbrush as an instrument for AR-based training and observed that repeated training in an AR helps users to continue performing the various activity in their actual living, thereby contributing to the development of proper habits. Further, this type of training helps users to enjoy participating in training, such as through games [8]. 

Most studies that applied tooth-brushing training based on AR using a smart toothbrush were conducted on school-aged children [13,19], with practically no studies examining people with intellectual disabilities. Therefore, this feasibility study aims to investigate the impact of AR-based training using a smart toothbrush on oral hygiene care among people with intellectual disabilities residing in residential care facilities in Korea. 

## 2. Materials and Methods

### 2.1. Study Subjects

The feasibility study was designed using a non-equivalent control group pre-test–post-test design. This study included 30 subjects residing in S residential care facilities for people with intellectual disabilities in I city of Korea, who agreed to participate after understanding the purpose and contents of this study. This study was approved by the Kangwon National University Institutional Review Board (IRB approval No.: KWNUIRB- 2019-09-006-003) and was conducted in accordance with the ethical standards of the Declaration of Helsinki. 

This study’s experiment consisted of tooth-brushing training based on the AR using a smart toothbrush (*n* = 15), and training based on the visual material (*n* = 15) (Figure 1). The inclusion criteria of the study subjects were as follows: First, they were diagnosed with sn intellectual disability based on the DSM-5. Second, they voluntarily agreed to participate in this study. The exclusion factors of the study subjects were as follows: First, they had neurological and orthopedic problems in their physical structures and function that disturbed psychomotor performance. Second, they received 20 points or less on the Korean Mini-Mental State Examination (K-MMSE), and they had difficulty understanding and conducting the test method [20]. Third, they were taking antipsychotic medications.

### 2.2. Tooth-Brushing Training Based on the AR Using Smart Toothbrush

For training based on AR using a smart toothbrush, Brush Monster (Kitten Planet Co., Yongin in Korea) was used. This is a smart toothbrush providing training based on AR that allows the user to learn proper tooth-brushing performance to form correct tooth-brushing habits. This smartbrush, using AR, can be directly applied to the user’s oral cavity, and it leads users to participate in tooth brushing pleasantly through game elements included in the AR of the toothbrush. In addition, as the 3D motion sensor in this brush enables sensitive recognition of the user’s motion, this brush provides a systematic and visual guide for tooth-brushing posture and direction. The size of the brush was 15 × 14.5 × 176.5 mm, and its weight was 32 g. It uses an antimicrobial head to avoid stimulating the gums, and is a sonic electric toothbrush with 1600 vibrations/minute [21].

Training begins at the time of moving the smart toothbrush in the user’s hands, and then guides users to brush their teeth while looking at themselves in AR on the application of mobile phones. If the training is finished, the smart toothbrush stops, and then the results of the tooth-brushing performance are presented in the tooth-brushing habit report in the application. Training was conducted twice a week for 12 weeks, and each session was performed for 30 minutes by an occupational therapist who was a researcher in this study. 

### 2.3. Tooth-Brushing Training Using Visual Material

Tooth-brushing training using a visual material was conducted using the educational material of the Korea Health Promotion Institute (KHPI) for oral hygiene improvement, based on the study by Kim and Lee [22]. In the process of applying the educational material, the material for kindergarten and lower grades of elementary school of KHPI was used in consideration of the lower cognitive and learning levels of people with intellectual disabilities. This consists of an education and practice process. The education process provides an explanation of the importance of tooth brushing and overall oral hygiene, and the practice process consists of education and practice on how to brush teeth properly, and is guided towards practice using a manual toothbrush. Based on the educational material of KHPI, tooth-brushing training was performed using a general toothbrush, the training was conducted twice a week for 12 weeks, and each session lasted 30 minutes, guided by an occupational therapist who was a researcher in this study.

### 2.4. Measures

#### 2.4.1. Tooth-Brushing Performance Assessment

In this study, the Korean ver.—Modified Barthel Index (K-MBI) was used to measure the tooth-brushing performance, and the measurements of tooth-brushing performance were performed based on the performance and time of tooth brushing. The K-MBI is an effective assessment for measuring performance level in activities of daily living. The measurement areas consisted of 11 areas (personal hygiene, bathing, feeding toileting, stair climbing, dressing, bowel control, bladder control, ambulation, wheelchair, and chair-bed transfer). The performance of each area was measured using a 5-point scale, from 1 (dependent level) to 5 points (independent level), by direct observation and interview with an occupational therapist. Based on the K-MBI, the activity of tooth brushing in the area of personal hygiene was measured. Thus, tooth-brushing performance was measured using a 5-point scale, and the total score ranged from 4 to 20 points, with higher scores indicating better performance. 

Moreover, to analyze the tooth-brushing performance accurately, the tooth-brushing performance on teeth was measured by dividing the areas and regions of teeth based on the study by Korpela et al. [23] (Table 1). Areas of the teeth are divided into outer (facial) and inner (lingual) surfaces, and the regions of the teeth are divided into frontal (incisors and canines) and back (molars) teeth [24]. 

For the measurement of tooth-brushing time, the tooth-brushing time from start to finish was measured using a stopwatch, and the results were recorded in seconds. All measurements of the performance of tooth brushing were performed every month after pre-test by direct observation from an occupational therapist, who was not a researcher in this study. The interrater reliability of the K-MBI was 0.93–0.98, and Cronbach’s α was 0.84 [24]. In the assessment modified in this study, Cronbach’s alpha of activities of turning water tap on, picking up the toothpaste and brush, opening toothpaste, and putting toothpaste on a toothbrush was 0.93. Regarding the activity of teeth-brushing on the outer surface, Cronbach’s alpha was 0.92, and that of the inner surface was 0.97. Cronbach’s alpha of the activities of rinsing mouth, washing mouth, arranging toothbrush, and putting toothbrush down was 0.97.

#### 2.4.2. Oral Hygiene Assessment

To measure oral hygiene, the simplified oral hygiene index (S-OHI) was used before and after training. S-OHI is an effective assessment to measure overall oral hygiene through measurement of food debris from the surface of teeth and the level of calculus on the teeth surface. The sub-items consisted of debris, calculus index, and total points. Assessment was performed on the following teeth in the following order: maxillary right first molar, maxillary right central incisor, maxillary left first molar, mandibular left first molar, mandibular left central incisor, and mandibular right first molar. The tooth surfaces assessed were: buccal of maxillary molars, lingual of mandibular molars, and labial of maxillary and mandibular incisors [25]. 

In the measurement of the debris index, each tooth was measured for the presence of remaining food debris and extrinsic strained teeth. Zero points indicate no debris or stain present; one point indicates that soft debris covering not more than one-third of the tooth surface was examined, or the presence of extrinsic stains without debris, regardless of the surface area covered; two points indicate that soft debris covers more than one-third but not more than two-thirds of the exposed tooth surface; three points indicate that soft debris covers more than two-thirds of the exposed tooth surface. In the measurement of the calculus index, each tooth was measured for the level of coating tartar. Zero points indicated that no calculus was present; one point indicates that supragingival calculus covering no more than one-third of the exposed tooth surface was examined; two points indicate that supra-gingival calculus covers more than one-third but no more than two-thirds of the exposed tooth surface, or the presence of individual flecks of sub-gingival calculus around the cervical portion of the tooth; three points indicate that the supra-gingival calculus covers more than two-thirds of the exposed tooth surface or a continuous heavy band of sub-gingival calculus around the cervical portion of the tooth [26].

All points for each tooth were added, and the number of teeth measured was divided into debris and calculus indices. Total points ranged from 0 to 6 points, each index was allocated into three points, and a higher score indicated a poor level of oral hygiene [25]. Measurements were conducted by direct observation, and the results of oral hygiene were recorded and analyzed by numerical values. 

### 2.5. Statistical Analysis

The study used a non-equivalent control group pre-test–post-test design. This study measured tooth-brushing performance and time and oral hygiene before the training application (baseline) and after the training (after 12 weeks) over the total 12 weeks of the intervention period. The Friedman test was used to determine the significant difference in tooth-brushing performance and time among each test-outcome variable within a group. Wilcoxon’s rank test was used to determine the significant difference in oral hygiene between pre-outcome and post-outcome variables within a group. The Mann–Whitney U test was used to determine the significant difference in all outcome variables between the two groups. Statistical significance was set at *p* < 0.05. All statistical analyses were conducted using the SPSS ver. 23.0 (IBM, New York, NY, USA). 

## 3. Results

### 3.1. Characteristics of Subjects

The Mann–Whitney U test was used to analyze differences between groups. The results showed that there were no significant differences between the groups for any of these variables (Table 2). According to the disability grading criteria of the Ministry of Health and Welfare in South Korea, 2 grade indicates a person with an IQ from 35 to 50 who can train simple behavior in daily life and who can perform simple work that does not require special skills with some supervision and help, 3 grade indicates a person with an IQ between 50 and 70 who can apply social and work rehabilitation. 

### 3.2. Results of Tooth-Brushing Performance

Based on the results of tooth-brushing performance within the experimental group, all the results for tooth-brushing performance and time showed a continuous increase after the training. According to the results of statistical verification, there was only a statistically significant difference in all the results on tooth-brushing performance and time in the experimental group (Table 3). In addition, after statistical verification of the difference between groups, there was no statistically significant difference between groups in any of the results of the pre-test. However, there were statistically significant differences between the groups showed in all the results after the training (Table 4). Thus, based on the results, the application of tooth-brushing training based on AR using a smart toothbrush was more effective than training based on visual material. 

### 3.3. Results of Oral Hygiene

Through the results of oral hygiene within the experimental group, the debris index, calculus index, and total score decreased after training. The results showed that the debris and calculus indices and total scores were significantly different in the experimental groups (Table 5). In addition, there were significant differences in all the results after training between the two groups (Table 6). Thus, this study’s results showed that the application of tooth-brushing training based on AR using a smart toothbrush was more effective than training based on the visual material in oral hygiene.

## 4. Discussion

Tooth brushing must be performed after every meal, or at least before going to bed and after breakfast [13]. The proper formation of tooth-brushing habits from learning is important for tooth brushing, for which repeated training and learning are required. Tooth brushing is learned by emulating parents’ behaviors in young childhood and is naturally established as a habit through repeated practice [27]. However, people with intellectual disabilities have little learning experience during the process of growth and development, and their low intelligence, cognitive ability, and learning ability make it challenging for them to learn tooth-brushing techniques and patterns. Therefore, they have difficulty continuing to perform this behavior in their daily lives and forming a habit [10]. 

Accordingly, residential facilities for people with intellectual disabilities continuously administer tooth-brushing training using visual materials to manage oral health and improve oral hygiene [5]. However, this training is conducted only as a one-time event by instructors working at each facility, and is limited in its improvement of tooth-brushing performance, as it only provides knowledge and information about tooth brushing. Therefore, it is difficult to learn the techniques of tooth brushing and maintain the effectiveness of learning [9]. Hence, in this study, to address the problems in tooth-brushing training using visual materials and improve the effectiveness of the tooth-brushing training for people with intellectual disabilities, tooth-brushing training based on AR using a smart toothbrush was applied. 

The results of the tooth-brushing performance and oral hygiene of the subjects after the training showed a significant difference between groups. Moreover, the results of statistical verification between groups showed statistically significant differences between the groups. Accordingly, tooth-brushing training based on AR using a smart toothbrush was effective in improving tooth-brushing performance and oral hygiene in people with intellectual disabilities, and is more effective than training using visual materials. Most studies support the results of this study [8,13,19]. Thus, this study has clinical significance in tooth-brushing training for people with intellectual disabilities. 

Specifically, based on the results of this study, we determined that tooth-brushing training based on AR using a smart toothbrush is more effective in improving tooth brushing performance by increasing the brushing accuracy on the areas and regions of teeth than the training using the visual material. The use of 3D motion sensors in a smart toothbrush is a strength of tooth-brushing training compared to conventional tooth-brushing training using visual materials [19].

In particular, 3D motion sensors in a smart toothbrush recognize a user’s movement. Upon recognizing this, it provides clues to immediately detect errors in the user’s tooth-brushing motion, and provides assistance and guidance to induce accurate tooth-brushing performance. This allows the user to learn the correct tooth-brushing technique and thoroughly brush each part of the tooth [13]. Kim et al. [13] reported that the use of 3D motion sensors contributes to the recognition and distinction of teeth areas, provides immediate assistance and signals to increase the accuracy of tooth brushing performance, and effectively promotes the learning of performance patterns. 

Moreover, Graetz et al. [19] reported that AR’s 3D motion sensor enables an increase in the user’s visual perception and attention to tooth-brushing movements, and effectively improves the accuracy of the performance. In the study by Faria et al. [8], they reported that the benefit of a 3D motion sensor in AR acts to recognize the user’s movements during tooth brushing and to improve tooth-brushing accuracy. These effectively help people with intellectual disabilities, with low awareness and concentration, to learn the correct tooth-brushing pattern and performance skills. Therefore, to improve the tooth-brushing performance of people with intellectual disabilities, the application of training based on AR using a smart toothbrush should be considered.

In the results of tooth-brushing time within the group, the increase in tooth-brushing time for each area and region on teeth was only shown in the experimental group after the tooth-brushing training based on AR using a smart brush, and statistically significant difference between groups showed. In the case of tooth-brushing time, tooth-brushing training based on AR using a smart toothbrush is more effective than training using visual material. Information on tooth-brushing motion detected by the 3D motion sensor is displayed in real-time on the application screen using visual material. Through these, the user can easily recognize the performance status, monitor it, correct errors, and cope with problems appropriately during tooth brushing [27]. In the process, various content attracts the users to enjoy brushing their teeth, such as playing games. Accordingly, the user’s interest, motivation, and active participation in tooth brushing is encouraged so that users can continue to perform tooth brushing [19]. Thus, based on these results, we judged that the tooth-brushing time in the experimental group was continuous and that training based on AR using a smart toothbrush was more effective than the training using visual material in this study.

Kim et al. [13] and Watt et al. [28] reported that tooth-brushing training based on AR using a smart toothbrush leads to an increase in tooth-brushing time on each tooth by motivating active participation in tooth brushing, and they reported that it is effective in promoting continuous participation. Graetz et al. [19] reported that tooth-brushing training based on AR is effective in increasing tooth-brushing time and is particularly effective in motivating users and increasing the level of commitment in tooth brushing. Hachisu et al. [29], Ostberg [30], and Flagg et al. [31], also stated that real-time monitoring and visual feedback in the training process using a smart toothbrush can identify the user’s performance status and correct errors in performance. Accordingly, they reported that the user’s tooth-brushing time increased effectively, and users continued to brush their teeth actively. 

Actually, the AR applied on the smart toothbrush induces the user to feel a sense of reality and immersion in tooth-brushing training, and the user effectively acquires self-control in the process of training [32]. Kallestal et al. [33] reported that tooth-brushing training based on AR using a smart toothbrush allows users to observe their movement on the display and control it. Accordingly, they reported that more attention is paid to the performance and that the tooth-brushing time increases. Based on these reports, to increase the tooth-brushing time of people with intellectual disabilities, the tooth-brushing training based on AR using a smart toothbrush should be considered. 

Dental plaque management is essential to prevent tooth decay, and proper tooth-brushing effectively prevents calculus index and plaque formation by removing food debris. In order to remove food debris and calculus effectively, most studies emphasize the importance of adequate and valid tooth-brushing motions while brushing teeth with toothbrushes. Li et al. [27] and Geurs [34] reported that tooth brushing effectively removes food debris and calculus from the tooth surface and maintains a clean oral environment. In this study, all the results of oral hygiene within the group showed an increase after tooth-brushing training based on AR using a smart toothbrush, and a statistically significant difference was observed between groups. 

The results showed that tooth-brushing training based on AR using a smart toothbrush is more effective in improving oral hygiene in people with intellectual disabilities than training using visual material. According to this study’s results, oral hygiene was improved effectively through the increase of the tooth-brushing performance level and time, and removing food debris and calculus significantly. Accordingly, we decided to directly contribute to the improvement in oral hygiene for people with intellectual disabilities. Therefore, to improve oral hygiene effectively, tooth-brushing training based on AR using a smart toothbrush should be considered for people with intellectual disabilities. 

One limitation of this feasibility study is that we had a small sample size, small scope of sampling, and a small number of study groups, which limits the generalizability of the findings. Based on our results, we propose increasing the number of study subjects and groups and conducting follow-up studies. In future studies, in the process of tooth-brushing training based on AR using a smart toothbrush, it is necessary to determine which factor contributes to the improvement in tooth-brushing performance and time through in-depth analysis. Moreover, the application of tooth-brushing training could also play a significant role in reducing the physical and psychological burden of caregivers and rehabilitation professionals. Thus, we propose conducting further studies on the satisfaction and burden of caregivers and rehabilitation professionals in relation to the application of tooth-brushing training based on AR using a smart toothbrush.

## 5. Conclusions

As a result, it was shown that tooth-brushing training based on AR using a smart toothbrush was more effective than the training using visual materials in improving oral hygiene care among people with intellectual disabilities. Tooth-brushing training based on AR using a smart toothbrush can be used as an effective tool to solve the problems of the training using visual material in the residential care facility for people with intellectual disabilities. Plus, it is effective to improve the performance of oral hygiene care by providing motivation for tooth brushing. Therefore, the tooth-brushing training based on AR using a smart toothbrush should be considered for application for people with intellectual disabilities residing in residential care facilities.

## Figures and Tables

**Figure 1 healthcare-09-00348-f001:**
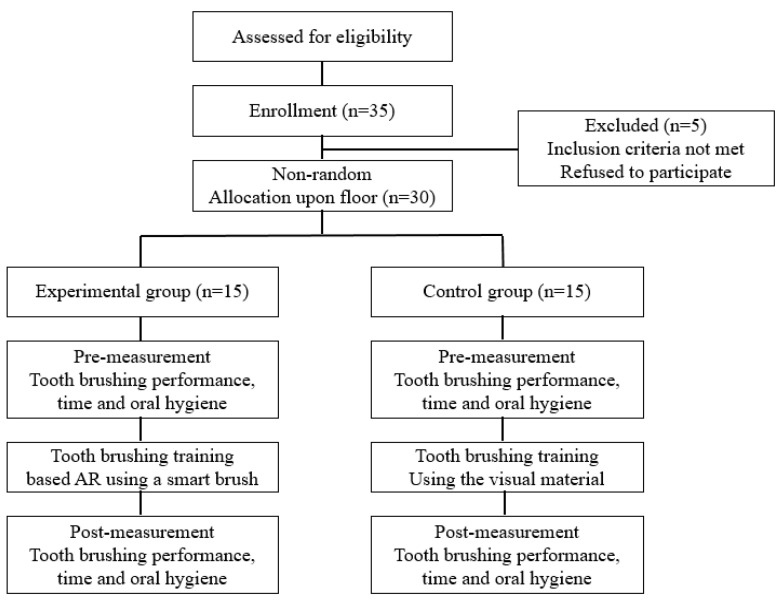
The schematic diagram of this study.

**Table 1 healthcare-09-00348-t001:** Tooth-brushing performance and time assessment.

Activities			Dependent	Maximal Assist	Partial Assist	Minimal Assist	Independent	Time	Cronbach’sAlpha
Performance of teeth brushing	Outer surface	Front teeth							0.919
		Back teeth							
	Inner surface	Front teeth							0.961
		Back teeth							
Total scores			/20 points					/s	

**Table 2 healthcare-09-00348-t002:** Characteristics of subjects.

Characteristics		Experimental Group	Control Group	U	Z	*p*-Value
Age (Mean ± S.D)		45.33 ± 10.42	45.67 ± 10.34	106.50	−0.250	0.806
Sex (*n*, %)	M	7 (46.7)	7 (46.7)	50.00	0.000	1.000
	F	8 (53.3)	7 (53.3)			
Disability grade (*n*, %)	2	10 (66.7)	10 (66.7)	50.00	0.000	1.000
	3	5 (33.3)	5 (33.3)			

**Table 3 healthcare-09-00348-t003:** The results of tooth-brushing performance and time within a group.

Groups	Tooth Brushing	Pre (Mean ± S.D)	1 Month after (Mean ± S.D)	2 Month after (Mean ± S.D)	3 Month after (Mean ± S.D)	Mean Difference	X^2^	*p*-Value
Experimental group	Performance (points)	12.27 ± 3.20	15.47 ± 2.64	17.20 ± 1.82	17.60 ± 1.84	5.33	34.756	0.000 ***
	Time (sec)	85.80 ± 9.73	89.67 ± 9.19	95.73 ± 9.52	101.13 ± 7.68	15.33	27.658	0.000 ***
Control group	Performance (points)	12.80 ± 3.69	12.40 ± 3.56	14.13 ± 2.88	12.73 ± 3.15	−0.07	3.000	0.392
	Time (sec)	84.07 ± 9.65	83.40 ± 9.92	87.47 ± 8.79	87.00 ± 8.49	2.93	3.857	0.277

*** *p* < 0.001; Friedman test.

**Table 4 healthcare-09-00348-t004:** The results of tooth-brushing performance and time between 2 groups.

Test	Tooth Brushing	Experimental Group (Mean ± S.D)	Control Group (Mean ± S.D)	Mean Difference	Z	*p*-Value
Pre	Performance (points)	12.27 ± 3.20	12.80 ± 3.69	0.53	−0.815	0.461
	Time (sec)	85.80 ± 9.73	84.07 ± 9.65	−1.73	−0.462	0.653
I Month after	Performance (points)	15.47 ± 2.64	12.40 ± 3.56	−3.07	−2.727	0.007 **
	Time (sec)	89.67 ± 9.19	83.40 ± 9.92	−6.27	−3.238	0.001 **
2 Month after	Performance (points)	17.20 ± 1.82	14.13 ± 2.88	−3.07	−3.216	0.001 **
	Time (sec)	95.73 ± 9.52	87.47 ± 8.79	−8.26	−2.861	0.004 **
3 Month after	Performance (points)	17.60 ± 1.84	12.73 ± 3.15	−4.87	−4.210	0.000 ***
	Time (sec)	101.13 ± 7.68	87.00 ± 8.49	−14.13	−3.688	0.000 ***

***p* < 0.01, ****p* < 0.001; Mann–Whitney U Test for the comparison analysis between 2 groups.

**Table 5 healthcare-09-00348-t005:** The results of oral hygiene within a group.

Groups	Items	Pre (Mean ± S.D)	Post (Mean ± S.D)	Mean Difference	Z	*p*-Value
Experimental group	Debris index (points)	1.32 ± 0.56	0.99 ± 0.38	−0.33	−2.673	0.008 **
	Calculus index (points)	1.49 ± 0.69	0.86 ± 0.51	−0.63	−2.940	0.003 **
	Total score (points)	2.81 ± 1.14	1.87 ± 0.86	−0.94	−2.940	0.003 **
Control group	Debris index (points)	1.45 ± 0.78	1.37 ± 0.37	−0.08	−0.089	0.929
	Calculus index (points)	1.33 ± 0.88	1.29 ± 0.43	−0.04	−0.569	0.569
	Total score (points)	2.78 ± 1.56	2.73 ± 0.81	−0.05	−0.408	0.683

** *p* < 0.01; Wilcoxon’s Ranked Test.

**Table 6 healthcare-09-00348-t006:** The results of oral hygiene between 2 groups.

Tests	Items	Experimental Group	Control Group	Mean Difference	Z	*p*-Value
Pre	Debris index (points)	1.32 ± 0.56	1.45 ± 0.78	0.13	−0.925	0.367
	Calculus index (points)	1.49 ± 0.69	1.33 ± 0.88	−0.16	−0.520	0.624
	Total score (points)	2.81 ± 1.14	2.78 ± 1.56	−0.03	−0.166	0.870
Post	Debris index (points)	0.99 ± 0.38	1.37 ± 0.37	0.38	−2.455	0.015 *
	Calculus index (points)	0.86 ± 0.51	1.29 ± 0.43	0.43	−2.020	0.045 *
	Total score (points)	1.87 ± 0.86	2.73 ± 0.81	0.86	−2.141	0.033 *

* *p* < 0.05; Mann-Whitney U test for comparison between two groups.

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
