# Peer review of "Effects of Tooth Brushing Training, Based on Augmented Reality Using a Smart Toothbrush, on Oral Hygiene Care among People with Intellectual Disability in Korea"

_healthcare, 2021, doi:10.3390/healthcare9030348_

Round 1

Reviewer 1 Report

This study explored the effect of AR based training using a smart toothbrush on tooth brushing performance, time, and oral hygiene of people with intellectual disability residing in the residential care facility in Korea. This study revealed that tooth brushing training based on AR using a smart tooth-brush was associated with oral hygiene care among people with intellectual disability. Here are my suggestions:

Title

Is it possible to shorten the title? For example, “Effects of tooth brushing training based on augmented reality using a smart toothbrush on oral hygiene care among people with intellectual disability in Korea.”

Abstract

In the abstract, AR may need to come with a full name when it was mentioned the first time.

Introduction

The introduction is concise and well written. It provides a clear rationale for the study, noting that a little knowledge regarding the effect of AR based training using a smart toothbrush on brushing performance and oral hygiene care among people with intellectual disability. It reasonably cites and satisfactorily discusses pertinent literature.

Methods

  1. The authors might briefly describe how they identified unperformed, incomplete performance, and normal performance in Line 147-148. Did they borrow the idea from previous literature?
  2. In line 181, Table 2 should be Table 1.

Results

The results are well presented with the five tables necessary, including use of nonparametric analyses.

  1. In the tables, I suggest the authors keep P-value to show the statistical significance. Also, the authors might add which tests they adopt in the table’s footnote.
  2. It is great if the authors could provide a mean difference in the Table3, 4, 5, and 6.
  3. In Table 4, please add the units for both performance and time.
  4. I suggest the authors may report Table 4 and 6 to demonstrate the effect of intervention because the same results were shown between Table 3 and 4 and Table 5 and 6. They can briefly mention the tests within each group in the footnote.

Discussion

The findings of this study revealed that people with intellectual disability who received tooth brushing training based on AR using a smart tooth-brush had better oral hygiene care. Overall, the discussion is reasonably written.

Conclusion

The conclusion is clearly stated and justifiable based on the analysis.

Author Response

Response to Reviewer 1 Comments

Point 1: Is it possible to shorten the title? For example, “Effects of tooth brushing training based on augmented reality using a smart toothbrush on oral hygiene care among people with intellectual disability in Korea.”

Response 1: Possible to shorten the title. Through the comment, the title was shorted “Effects of tooth brushing training based on augmented reality using a smart toothbrush on the tooth brushing performance and oral hygiene among people with intellectual disability in Korea.

Point 2: In the abstract, AR may need to come with a full name when it was mentioned at the first time.

Response 2: In the abstract, according to the comment, the AR modified to augmented reality.

Point 3: The introduction is concise and well written. It provides a clear rationale for the study, noting that a little knowledge regarding the effect of AR based training using a smart toothbrush on brushing performance and oral hygiene care among people with intellectual disability. It reasonably cites and satisfactorily discusses pertinent literature.

Response 3: Thank you for your positive review. We tried to prove a clear basis for the research and to develop the instruction by citing related literature.

Point 4: The authors might briefly describe how they identified unperformed, incomplete performance, and normal performance in Line 147-148. Did they borrow the idea from previous literature?

Response 4: The contents presented are the contents displayed by the application connected the smart brush. However, this was deleted as it was not actually used in the results section of this study.

Point 5: In line 181, Table 2 should be Table 1.

Response 5: According to the comment, Table 2 was modified to Table 1.

Point 6: In the tables, I suggest the authors keep P-value to show the statistical significance. Also, the authors might add which tests they adopt in the table’s footnote.

Response 6: In the tables, we keep P-value to show the statistical significance clearly. However, through the adding the footnote, the tables might be more complicated, so, the footnote should not be added.

Point 7: It is great if the authors could provide a mean difference in the Table3, 4, 5, and 6.

Response 7: In the case of table 3, the mean difference only reported between pre and 3 month after and, this showed only overall changes. Additionally, as the results of repeated measurements showed, if the mean difference will be added, the table should be more complicated and the l am concerned that it might undermine readers’ understanding. Thus, l think it is better not to write the mean difference.

Point 8: In Table 4, please add the units for both performance and time.

Response 8: According to the comment, in Table 4 and 6, the units added.

Point 9: I suggest the authors may report Table 4 and 6 to demonstrate the effect of intervention because the same results were shown between Table 3 and 4 and Table 5 and 6. They can briefly mention the tests within each group in the footnote.

Response 9: Thank you for your positive review. I agree your comment that it is more important results in table 4 and 6 than table 3 and 5. However, l think that the results of the changes in each group might be important, so l think it is better to keep the table 4 and 6 , not to unremoved.

Point 10: The findings of this study revealed that people with intellectual disability who received tooth brushing training based on AR using a smart tooth-brush had better oral hygiene care. Overall, the discussion is reasonably written

Response 10: Thank you for your positive review. We tried to state reasonably in the discussion section about this study.

Point 11: The conclusion is clearly stated and justifiable based on the analysis

Response 11: Thank you for your positive review. We tried to state clearly in the conclusion section about this study.

Reviewer 2 Report

Authors aimed to identify the effects of tooth brushing training with a smart toothbrush measured by accuracy of performance, time and quality of oral hygiene in a sample of 15 subjects with intellectual disabilities. These effects were compared with a control group of 15 similar subjects using standard visual aids. Results found significant differences between the experimental and control groups on accuracy of performance, time of brushing and quality of oral hygiene.

I will first comment on the overall quality of the manuscript. The manuscript definitely suffers from poor knowledge of the English language. It was a struggle to wade through very lengthy and repetitive descriptions and maintain a sense of understanding about what authors were presenting. Also, the research does not seem especially original, as the authors themselves point out in their bibliographic literature comparisons. If the manuscript is to be revised for this or other journals, I enclose a pdf with specific markings and comment bubbles to aid in that process, as well as an attempt to exemplify language suggestions. However, this was not comprehensive, since language problems were so extensive. An English speaking writer needs to be consulted to improve author formulations, shorten the text and do away with repetitions.

The statistical analysis seemed appropriate to support the results and discussion.

The schematic diagram of the study appears in error, since descriptions of the experimental and controls are the same as if they were copy-pasted in error.

The discussion was much too lengthy and repetitious. I have indicated in the pdf some cuts that could be made (see especially pg 11-12), although cut suggestions I make are not exhaustive. There were also contradictions: e.g. lines 363-365 contradict lines 392-397. And findings, therefore, are not very clear related to time of brushing as positive or negative.

All in all, this manuscript is of poor quality and would require major revision to even be considered again for publication. The authors must assess if they would be willing to put all the needed work into it and get an English consultant.

Author Response

Point 1: I will first comment on the overall quality of the manuscript. The manuscript definitely suffers from poor knowledge of the English language. It was a struggle to wade through very lengthy and repetitive descriptions and maintain a sense of understanding about what authors were presenting. Also, the research does not seem especially original, as the authors themselves point out in their bibliographic literature comparisons. If the manuscript is to be revised for this or other journals, I enclose a pdf with specific markings and comment bubbles to aid in that process, as well as an attempt to exemplify language suggestions. However, this was not comprehensive, since language problems were so extensive. An English speaking writer needs to be consulted to improve author formulations, shorten the text and do away with repetitions.

Response 1: According to the comment about the quality of this manuscript, the detailed review should be performed by all authors carefully. And the English correction also was done in the revision process to improve the quality of the expression in this study.

Point 2: The statistical analysis seemed appropriate to support the results and discussion.

Response 2: Thank you for your positive review.

Point 3: The schematic diagram of the study appears in error, since descriptions of the experimental and controls are the same as if they were copy-pasted in error.

Response 3: The schematic diagram of the study modified according to this study process.

Point 4: The discussion was much too lengthy and repetitious. I have indicated in the pdf some cuts that could be made (see especially pg 11-12), although cut suggestions I make are not exhaustive. There were also contradictions: e.g. lines 363-365 contradict lines 392-397. And findings, therefore, are not very clear related to time of brushing as positive or negative.

Response 4: According to the comment, we reduced the length of discussion and the repetition in discussion. Also, the contents included in line 392-397 means the time from preparation of tooth brushing to arrangement after tooth brushing based on the Table 2. Thus, is it right the increase the time of teeth brushing (each teeth) and decrease the total time of tooth brushing including the preparation and arrangement of tooth brushing. However, in this study, the results of the teeth brushing time in more important than the decrease the total time of tooth brushing, the overall contents related in the total time of tooth brushing should be removed. Plus, according to the comment in the pdf file, the contents should be modified carefully.  

Point 5: All in all, this manuscript is of poor quality and would require major revision to even be considered again for publication. The authors must assess if they would be willing to put all the needed work into it and get an English consultant.

Response 5: Like the response 1, the careful modification should be performed for the improvement of quality of the manuscript contents and expression in English.

In the pdf file

Points: appears to be .. future research needed to confirm.

Results: The results of the performance and time of tooth brushing were increased continuously after the training and the results of oral hygiene were improved in experimental group only

Response: The contents modified as follows: “All the results of the tooth brushing performance and oral hygiene showed the increase in the experimental group only. In addition, there were significantly different in all results after training between 2 groups”.

Points: should be needed -> is needed

Response: Modification is done from “should be needed” to “is needed”.

Points: falls under -> is a

Response: Modification is don from “falls under” to “is a”.

Points: and it is a highly -> that is

Response: Modification is done from “and it is a highly” to “That is”.

Points: activity that -> for

Response: Modification is done from” activity that” to “for”.

Points: Impact -> Impacting

Response: Modification is done from “impact” to “impacting”.

Points: problems added in line 51

Response: Added the problems in line 50.

Points: what about electric toothbrush.. why not discuss?

Points: again, why no discussion of electric toothbrushing?

Response: This study purpose is to analyse the effect of the training based on AR using a smart toothbrush. Also, general toothbrushes and electric toothbrushes are the same in terms of learning to perform toothbrushes, and only the strength of the stimuli generated by the toothbrushes is different. I think the users both tooth brushes to need the additional brushing training because it is the same to form a proper brushing habit. Therefore, we did not include the discussion of electric toothbrushes, but the discussion of smart toothbrushes was presented in the next paragraph.

Points: poor English

Response: The contents were modified to express clearly from the line 62 to 68.

Points: persisted -> persistent

Response: Modification is done from “persisted” to “persistent”.

Points: This needs to be written in the abstract as well.

Response: This sentence moved to the abstract section. “In the modern society, augmented reality based training using a smart device has emerged as a means to resolve such problems with training.”

Points: numeral -> numeric

Response: Modification is done from “numeral” to “numeric”.

Points: What is this? ADL

Response: ADL indicates activities of daily living. But, these contents is removed and the sentence was modified to “There were also multiple studies that applied AR based training for tooth brushing, which is the activities of daily living essential to managing oral hygiene”.

Points: ??

Response: Residential care facility indicate the long term rehabilitation facility for the people with intellectual disability based home in Korea (located in the community, not group home facility).

Points psychomotor added

Response: “psychomotor” added in line 119.

Points: is -> was

Response: Modification is done from “is” to “was” in line 249 and 257.

Point: in line 277, a add

Response: in line 277, “a” added.

Points : rambles too much, get to the points

Response: According to the comment, we tried to reduce the length and state the points clearly.

In addition to this, all the corrections presented in the pdf file have been completely revised, and the parts that need to be deleted have been revised and marked in the text.

    Thank you for the detailed review.

Reviewer 3 Report

Introduction

Line 38 - “… increase oral malodor…” – The authors must use the scientific term

Methods and Materials

The number of groups seems to me to be too small to draw consequent conclusions, since patients do not have a pattern of intellectual disability exactly the same.

119-126 - This entire paragraph refers to inclusion / exclusion factors. Thus, the authors should use this terminology.

Fig 1. This figure has several errors. It uses the same criteria in the control group and in the experimental group.

For the control group only visual methods were used. The control group should have been teached to brush their teeth with a manual brush so a comparison between a manual and a smart brush could be made. How can the authors say the smart brush is better if no manual brush was used as control?

References

References are generally very old. Half of the references were published before 2010 and two articles were from 1968 and 1964 respectively.

Author Response

Response to Reviewer 3 Comments

Point 1: Line 38 - “… increase oral malodor…” – The authors must use the scientific term

Response 1: According to the comment, changes the word from “the oral malodour” to “oral halitosis”

Point 2: The number of groups seems to me to be too small to draw consequent conclusions, since patients do not have a pattern of intellectual disability exactly the same.

Response 2: I already know about that. So, these contents was wrote in the limitation section and the more studies which expand the number of study subjects should be conducted to verify the changes or effect of the tooth brushing training based on AR using a smart toothbrush.

Point 3: 119-126 - This entire paragraph refers to inclusion / exclusion factors. Thus, the authors should use this terminology

Response 3: The terminology (inclusion / exclusion) added in the contents.

Point 4: Fig 1. This figure has several errors. It uses the same criteria in the control group and in the experimental group

Response 4: I catch the error so the error was modified.

Point 5: For the control group only visual methods were used. The control group should have been teached to brush their teeth with a manual brush so a comparison between a manual and a smart brush could be made. How can the authors say the smart brush is better if no manual brush was used as control?

Response 5: in the tooth brushing training using visual material, the use of general toothbrush was already stated. However, to confirm the

Point 6: References are generally very old. Half of the references were published before 2010 and two articles were from 1968 and 1964 respectively.

Response 6: I know that references are very old, but they are very important references and are always cited as manuals on related topics. I know that references are very old, but they are very important references and are always cited as manuals on related topics.

Round 2

Reviewer 3 Report

References

References are generally very old. Half of the references were published before 2010 and two articles were from 1968 and 1964 respectively.

- It is proposed that the authors increase the n of the groups in order to drawn conclusion more consistently.

Author Response

Point 1: Line 38 - “… increase oral malodor…” – The authors must use the scientific term

Response 1: According to the comment, changes the word from “the oral malodour” to “oral halitosis”

Points 2: The number of groups seems to me to be too small to draw consequent conclusions, since patients do not have a pattern of intellectual disability exactly the same.

Response 2: This contents is described in the study limitation part, and the limitation about the number of study groups is also described.

Points 3: 119-126 - This entire paragraph refers to inclusion / exclusion factors. Thus, the authors should use this terminology.

Response 3: the term of inclusion / exclusion factors were used.

Points 4: Fig 1. This figure has several errors. It uses the same criteria in the control group and in the experimental group.

Response 4: In the figure, the errors were modified as the figure in the manuscript.

Points 5: For the control group only visual methods were used. The control group should have been teached to brush their teeth with a manual brush so a comparison between a manual and a smart brush could be made. How can the authors say the smart brush is better if no manual brush was used as control?

Response 5: The clear purpose of this study is to compare brushing training methods.

It is using a smart brush to apply brushing training using augmented reality. The smart brush applied in the experimental group of this study is a toothbrush in the form of a general toothbrush that does not provide electrical stimulation like an electric toothbrush. However, it has a built-in sensor so it can recognize the motion of the person using the toothbrush. These contents are described in the research method, and the contents have been summarized in rder to present them more clearly.(in the tooth brushing using visual material).

Points 6: References are generally very old. Half of the references were published before 2010 and two articles were from 1968 and 1964 respectively.

Response 6: The references were modified according to the comment.
